# Liquid Biopsies in Lung Cancer

**DOI:** 10.3390/cancers15051430

**Published:** 2023-02-23

**Authors:** Marcel Kemper, Carolin Krekeler, Kerstin Menck, Georg Lenz, Georg Evers, Arik Bernard Schulze, Annalen Bleckmann

**Affiliations:** 1Department of Medicine A for Hematology, Oncology and Pneumology, University Hospital Muenster, 48149 Muenster, Germany; 2West German Cancer Center, University Hospital Muenster, 48149 Muenster, Germany

**Keywords:** liquid biopsy, lung cancer, ctDNA, CTC, miRNA, EV

## Abstract

**Simple Summary:**

Liquid biopsy has recently been introduced as a novel method in cancer diagnostics. It is less invasive for patients than conventional tissue biopsies, as the assay material is drawn from peripheral blood. The detected DNA fragments, as well as the cells and extracellular vesicles, can be used as biomarkers for cancer. Not all biomarkers are equally reliable in cancer diagnostics. In this review, we give an overview of the lung cancer biomarkers identified in liquid biopsy assays and discuss the differences, current applications, and future perspectives of liquid biopsies in lung cancer.

**Abstract:**

As lung cancer has the highest cancer-specific mortality rates worldwide, there is an urgent need for new therapeutic and diagnostic approaches to detect early-stage tumors and to monitor their response to the therapy. In addition to the well-established tissue biopsy analysis, liquid-biopsy-based assays may evolve as an important diagnostic tool. The analysis of circulating tumor DNA (ctDNA) is the most established method, followed by other methods such as the analysis of circulating tumor cells (CTCs), microRNAs (miRNAs), and extracellular vesicles (EVs). Both PCR- and NGS-based assays are used for the mutational assessment of lung cancer, including the most frequent driver mutations. However, ctDNA analysis might also play a role in monitoring the efficacy of immunotherapy and its recent accomplishments in the landscape of state-of-the-art lung cancer therapy. Despite the promising aspects of liquid-biopsy-based assays, there are some limitations regarding their sensitivity (risk of false-negative results) and specificity (interpretation of false-positive results). Hence, further studies are needed to evaluate the usefulness of liquid biopsies for lung cancer. Liquid-biopsy-based assays might be integrated into the diagnostic guidelines for lung cancer as a tool to complement conventional tissue sampling.

## 1. Introduction

In recent years, lung cancer has become the second most frequent, newly diagnosed malignant disease worldwide, and is the main cause of cancer-specific mortality [1]. The timely detection of these tumors is difficult, as early symptoms, such as cough or dyspnea, are unspecific or absent [2], and because current screening methods, such as low dose computed tomography (LD-CT), albeit effective, are not available on a large scale. Managing the tumor patient and adapting the therapy can be challenging due to the fact that lung cancer is a very heterogeneous disease, even in a single patient. For instance, PD-L1 expression can differ between the primary tumor, lymph node, and metastatic site tissue [3]. Currently, the standard sampling of suspected tumor tissue is resource consuming, painful, and potentially dangerous for the patient. In addition, the obtained material only represents the tissue at the specific biopsy site, and thus can entirely overlook the aspect of tumor heterogeneity, which is essential for effective therapy. This is especially true in patients undergoing targeted treatment of driver mutations as the treatment-induced selection of drug resistant cells [4] with the subsequent progression of such being a major prognostic issue. Progressive lesions in the CT scan might call for a therapeutic switch or treatment beyond disease progression (TBP), that also might be beneficial in immunotherapeutic or targeted therapy settings [5,6]. However, such decisions would be hard to make without definitive data on the actual makeup of the tumor.

In this setting, the liquid-biopsy-based assay, or simply the liquid biopsy, a method for detecting, identifying, and quantifying DNA and cell fragments in body fluids, can be an important addition to the palette of methods for the diagnosis and management of lung cancer patients. The method itself has a long history—scientists first detected cell-free DNA (cfDNA) in blood in 1948 [7]. Others were able to identify specific mutations of cfDNA in 1994, but it wasn’t until 2016 that the FDA approved the first liquid biopsy test for detecting EGFR gene mutations in lung cancer. The foremost value of a liquid biopsy is that one can obtain a synopsis of the relevant tumor information, not only of the known tumors, but also of any yet undetected metastatic tumors in the body, by analyzing the sample for circulating tumor DNA (ctDNA).

In brief, samples of peripheral blood are analyzed for cfDNA (150–200 base pairs [bp]) and ctDNA (90–150 bp), as well as for microRNA (miRNA), which directly originate from apoptotic or necrotic cells or are released from tumor-associated macrophages (TAMs). Circulating tumor cells (CTCs) may also be found in the blood samples, but the information from the circulating DNA fragments is more revealing than that from single tumor cells, since it represents the heterogeneity of all shed cells, including NSCLC cells at a specific time point.

As mentioned previously, cfDNA consists of small DNA fragments that are either passively released from apoptotic or necrotic cells or released by phagocytes due to digestion [8]. ctDNA is a subentity of cfDNA that is shed by malignant cells [9]. Apoptosis promotes protein-bound cfDNA and ctDNA in peripheral blood, also known as oligo- or mono-nucleosomes. However, ctDNA can also be found in membrane-bearing EVs that are actively released by cells [10].

In the current guidelines for the management of patients with non-curatively treated lung cancer, the liquid biopsy has found its way into the routine diagnostics for EGFR-mutated tumors, once the disease has progressed under a non-T790 M-addressing EGFR tyrosine kinase inhibitor [11]. Due to the fact that the concentration of fragmented DNA in the blood plasma of cancer patients is orders of magnitude higher than that in non-cancer controls (5–1500 ng/mL vs. 1–5 ng/mL), it can be used as a diagnostic tool [12]. A liquid biopsy offers an altogether minimally invasive method for monitoring treatment responses (i.e., minimal residual disease [MRD] monitoring), and helps us to understand genomic tumor evolution and to identify resistance mutations earlier than is possible with radiological findings.

## 2. Methodology (Figure 1) 

The materials and methods currently used in liquid and tissue biopsies for diagnostic assessments of lung cancer can be found in Figure 1.

**Figure 1 cancers-15-01430-f001:**
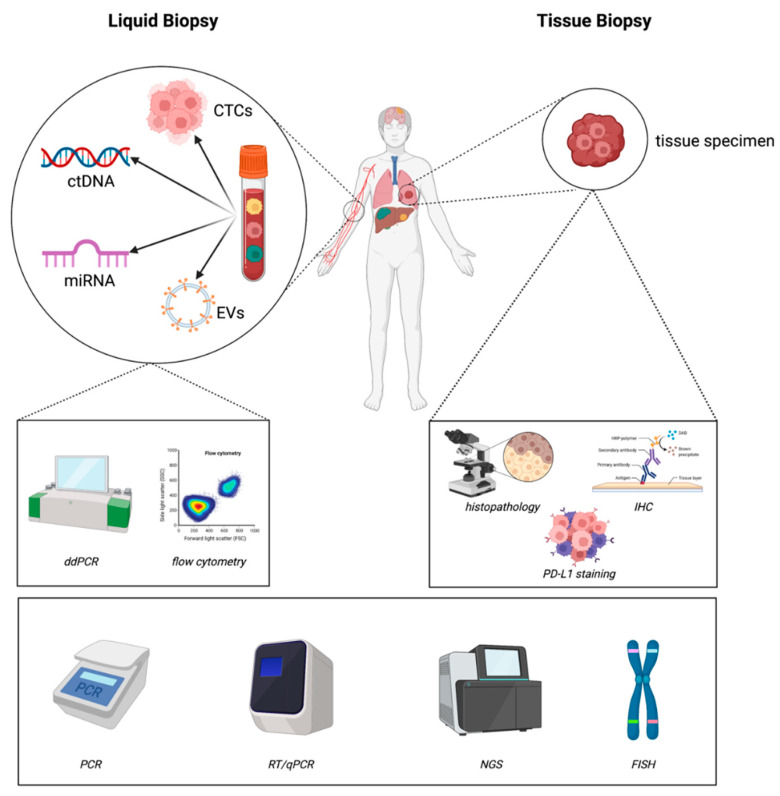
Materials and methods currently used in liquid and tissue biopsies for diagnostic assessments of lung cancer (created with https://biorender.com/ accessed on 22 February 2023).

### 2.1. Cell Free DNA (cfDNA) and Circulating Tumor DNA (ctDNA)

#### 2.1.1. Sample Collection and Preparation Timeline

Although the cfDNA content is higher in serum than in plasma [13], plasma is the preferred source for extraction, since there is less DNA contamination due to clotting, cell lysis, and the release of cfDNA from white blood cells [14]. A standard volume for a sample collection has not yet been defined, but 20 mL blood samples are usually employed for a cfDNA analysis in most isolation protocols [14]. 

The detection and identification of ctDNA is the preferred method of the liquid biopsy for lung cancer. As the levels of ctDNA in the plasma of NSCLC patients are very low (<0.5% of total cfDNA) [15], and it has only a short half-life < 3 h [16], time-sensitive isolation protocols should be implemented. The procedural steps in isolating ctDNA are: (1) drawing blood, (2) centrifugation, (3) DNA extraction, and (4) DNA analysis. 

#### 2.1.2. Collecting Tubes

Blood samples can be collected in either standard EDTA tubes or in specialized tubes containing a cfDNA preservative (see Table 1). EDTA tubes are widely available and considerably less inexpensive than the specialized tubes with the preservative. Disadvantageously, the samples from EDTA tubes have to be processed within 1–2 h to avoid their contamination with cfDNA from leukocyte lysis [17], but the samples from the tubes containing the cfDNA preservative can be stored at room temperature for up to 14 days [18]. However, for cfDNA preservative tubes, the contamination with leucocyte DNA can commence earlier so that processing should begin within three days [14]. The disadvantage with these tubes is that they require special processing and must often be sent to external laboratories, incurring delays and further costs.

Even in immediately processed samples, tumor-specific somatic mutations often represent < 1% of the total cfDNA for the region of interest, and any increase in the contamination with “diluting” cfDNA from in vitro lysis may result in false-negative results [19]. Consequently, a two-step centrifugation of the EDTA samples is recommended to reduce the contamination from leukocyte DNA, prior to their freezing and storage [19].

**Table 1 cancers-15-01430-t001:** Comparison of blood collection tubes.

	EDTA Tubes	Preservative Tubes
Advantages	Inexpensive and easily available Broad experience with use Better than heparin or citrate for anticoagulation [20] Transport to external laboratory not required	PAXgene^®^ Blood DNA tubes (Qiagen) or Cell-free DNA BCT^®^ tubes (Streck) cfDNA stable up to 14 days and CTC up to 7 days [18] Can be stored at room temperature [21]
Disadvantages	Must be processed within 1–2 h to prevent contamination with DNA from white blood cells	Expensive Limited availability and experience Strict adherence to product guidelines necessary Transportation to external laboratory required

#### 2.1.3. Methods for DNA Analysis

In total, two established methods are available for analyzing nucleic acids (e.g., DNA and RNA): targeted polymerase chain reaction (PCR)-based, and next generation sequencing (NGS)-based assays. Table 2 highlights the differences between the methods. 

#### 2.1.4. Real-Time (quantitative) PCR (qPCR) Assay

A qPCR is a widely used method for DNA analysis, which also allows for the semi-quantification of samples. The cost of the method has decreased markedly, and the results are easy to interpret. There are several commercially available FDA- and EMA-approved tests for detecting, activating, or resistance-causing EGFR mutations in NSCLC. Because only a single gene or an already known genetic alteration can be examined, the sensitivity is only 70–80% (detection limit 1–5%) [23,24], and using a qPCR alone carries the risk of false-negative results. 

#### 2.1.5. Digital Droplet PCR (ddPCR) Assay

A ddPCR is more sensitive than a qPCR (detection limit 0.1–1%) [23]. In ddPCR, a sample oil emulsion is fractionated into many thousands of droplets, upon which a PCR is subsequently performed in the individual compartments of a microtiter plate. This method allows for the detection of even rare events and quantification at the level of a single molecule. BEAMing (beads, emulsions, amplifications, and magnetics) has further improved the ddPCR technique using DNA templates that are bound to magnetic beads [25]. 

#### 2.1.6. Next Generation Sequencing (NGS) Assay

NGS is a high-throughput method for sequencing DNA and RNA that allows for the detection of single nucleotide polymorphisms (SNPs), and small (insertions and deletions) as well as large (insertions, deletions, amplifications, inversions, and translocations) genetic alterations. The method is very sensitive (detection limit 0.001–2%) and can detect even unknown genetic alterations in small DNA fragments [22]. Whole genome sequencing (WGS) and whole exome sequencing (WES) have not been established, as the amount of ctDNA obtained with liquid biopsies might be too low. So far, only hybrid capture-based and amplicon-based (PCR capture) NGS approaches are regularly used in routine testing for cancer diagnostics.

##### Hybrid Capture-Based NGS

The hybrid capture-based NGS approach is not based on primary amplification, and thus allows for a more reliable quantification of copy numbers than amplicon-based NGS. Targeted DNA sequences are hybridized (“captured”) to biotinylated probes, which are bound to magnetic beads. The beads are captured by magnets, and the non-hybridized DNA is washed off. Since the ctDNA plasma concentration is very low, and sequencing does not rely on prior amplification, this method is at a greater risk of sequencing errors, including false-positive results. This explains its low specificity (approx. 60%) [12]. 

##### Amplicon-Based NGS (PCR Capture)

The amplicon-based NGS approach is based on the primary PCR amplification of specific genomic regions of interest, especially hotspot genes. Before sequencing, the amplified DNA sequences (“amplicons”) are multiplexed and marked with a distinct molecular barcode for identification. As this approach uses primary PCR amplification, it is valuable for liquid biopsies with their low amounts of ctDNA [26]. The risk of false-positive results is effectively reduced using molecular barcodes. While amplification biases the quantification of allele frequencies and copy number variations (CNVs), this method is useful for the detection of SNPs, indels, or known gene fusions [12].

#### 2.1.7. Comparison of PCR-Based and NGS-Based Methods

PCR-based methods (qPCR, ddPCR) are well established, require short turnaround times of 2–3 days, and are fairly inexpensive. However, they allow for the detection of only a limited number of genetic alterations at a time (e.g., no multiplexing across different genes) and detect neither previously unknown alterations, nor gene fusions. NGS-based methods, on the other hand, have a longer turnaround time of about 1–2 weeks [27], but are able to detect known and unknown genetic variants, including CNVs, SNPs, and gene fusions, which might promote the use of diagnostic algorithms for cancer diagnostics and treatment monitoring. Lately, the NGS of *ALK*-positive NSCLC has offered a more detailed characterization of fusion partners than standard pathological techniques like IHC or FISH [28]. For quantification analysis, NGS and ddPCR are more suitable than the semi-quantitative qPCR. Despite the high sensitivity of NGS-based methods, a single false-positive read (with an error rate of approximately 10^−3^) can impact the result [29]. Hence, error-proofing techniques and algorithms need to be implemented.

### 2.2. Circulating Tumor Cells (CTCs)

Circulating tumor cells are rare in peripheral blood (1 CTC per 10^6−7^ leukocytes), where they occur as single tumor cells in cell clusters, as so-called circulating tumor microemboli [30], or attached to stromal cells originating from the primary tumor [31]. Their plasma half-life (1–2.4 h) is short, so that they must be promptly separated from other blood cells [32]. CTCs can be isolated by using antibodies to identify the expression of intracellular (DNA) or transmembrane molecules (cytokeratins, epithelial adhesion molecules [EGDR and HER3], and CD45) [33]. They can also be identified according to their density or size [12]. Circulating epithelial cells are present in only 46% of blood samples from stage IV NSCLC patients, and circulating cells with an epithelial phenotype can also be found in 7% of healthy controls [34]. Thus, the presence of CTCs in the bloodstream alone is an inadequate marker for the tumor burden. However, some studies have shown that the baseline CTC count is a prognostic marker for tumors other than lung cancer [35,36,37]. 

#### 2.2.1. Antigen-Based CTC Isolation

In the CellSearch^®^ CTC Test (Menarini Silicon Biosystems Inc., Castel Maggiore, Italy) the CTCs are separated magnetically from other blood cells using anti-EpCAM antibodies conjugated with magnetic nanoparticles. The separated cells are stained for nuclear DNA with 4′,6-diamidino-2-phenylindol (DAPI), and with fluorescently labeled antibodies for cytokeratins (CK) and CD45. They are then analyzed with automated fluorescence microscopy. CTCs are defined as DAPI^+^, CK^+^, and CD45^−^, while leukocytes are DAPI^+^ and CD45^+^ [33]. While this widely used assay is considered the gold standard for CTC detection, it has, so far, only been approved by the FDA for routine use in metastatic breast, prostate, and colorectal cancer, but not yet in lung cancer.

A different commercially available test (AdnaTest^®^ (Qiagen GmbH, Hilden, Germany)) uses a combination of different antibodies (e.g., anti-EpCAM, anti-MUC1, anti-HER2, and anti-EGFR) conjugated with magnetic beads for cell separation, which increases the sensitivity of the CTC detection when only the antibodies against EpCAM are used [38]. However, this test has not yet been approved by the FDA for routine use in cancer.

#### 2.2.2. CTC Isolation Based on Biological and Physical Characteristics

Antigen-based isolation methods have a poor sensitivity and specificity [31] due to the cells’ tendency for epithelial-to-mesenchymal transition (EMT) [39], with a subsequent loss of epithelial surface markers [40]. Focusing on the physical and biological characteristics of the CTCs, instead of the epithelial cell surface antigens, would bypass this problem. 

CTCs can also be isolated by using the physical properties of the cells, i.e., size and density. One method employs a porous membrane (pore diameter 8 μm) (e.g., ISET^®^ system,(Rarecells Diagnostics SAS, Paris, France)) to isolate the CTCs. Here, leukocytes are not totally eliminated. Thus, a subsequent further cell characterization with a cytomorphological or immunocytochemical analysis is required [41,42]. Another method is to centrifuge the plasma in a density gradient, e.g., a Ficoll-Paque^®^ solution, to separate the mononuclear cells (including the CTCs) from other blood cells (e.g., OncoQuick^®^ system, (Greiner BioOne International GmbH, Kremsmuenster, Austria)). The method is inexpensive but has a high rate of contamination with leukocytes [43,44]. 

Microfluidic technologies have recently evolved as a very appealing approach to CTC isolation [45]. Exemplarily, one such “CTC-chip” uses anti-EpCAM-coated microposts under precisely controlled laminar flow conditions to capture the CTCs [46], while the vortex technology uses microscale vortices to isolate the CTCs based on their physical characteristics, such as size or compressibility [12,47,48]. These approaches are antigen-independent, resulting in a high sensitivity (a low risk of false-negative results due to EMT) and high purity of the CTC isolates [12].

#### 2.2.3. Interpretation of CTCs

Apart from using the common morphological and biological features of malignant cells (e.g., nucleus size and chromatin structure) to discriminate the CTCs from other blood cells, immunocytochemistry can be used for CTC identification. Moreover, genomic analyses, such as DNA analysis, as well as RNA analysis (qPCR or RNA sequencing), can be performed on CTCs, and may reflect the tumor heterogeneity [33]. So far, proteomics using mass spectrometry and immunoblotting have not been implemented in CTC analysis. 

### 2.3. miRNA

MicroRNAs (miRNAs) are noncoding RNAs involved in the posttranscriptional regulation of gene expression that can act as tumor suppressors or oncogenes. Cell-free miRNA can be detected in various body fluids. It is passively released during cell lysis (e.g., necrosis and apoptosis) and actively secreted by cells for intercellular communication [49]. miRNAs are usually packed into EVs (see below), or coupled with Argonaute2 (Ago2) protein [50] or high-density lipoprotein (HDL) [51], and dispersed into the extracellular environment. A genomic analysis can be performed using the same methods as those for cfDNA (see previous section).

### 2.4. Extracellular Vesicles (EVs)

EVs are small membrane particles that are shed by all living cells and mediate intercellular communication by carrying proteins, lipids, and nucleic acids (e.g., DNA, RNA) from the secreting to the surrounding cells. A total of two distinct EV populations have so far been described: small EVs (sEVs, also called exosomes) with a diameter between 50–150 nm, and larger microvesicles (intermediate size vesicles, IEVs) with a diameter between 100–1000 nm [52]. sEVs are formed inside the cell through the inward budding of endosomal membranes, while IEVs bud directly from the plasma membrane [53]. Numerous studies have demonstrated that tumor-derived EVs are crucial in establishing a favorable tumor microenvironment, and thus pave the way for metastatic spread [54]. The large amount of EVs shed by tumor cells into the blood and other body fluids has opened up a new perspective on using EVs as cancer biomarkers in liquid biopsies, particularly in lung cancer. Due to their small size, the isolation and analysis of sEVs is time-consuming, and thus not suitable for clinical routine. In contrast, the extraction and analysis of IEVs with flow cytometry is less demanding and time-consuming [52]. Although EVs can be isolated from body fluids [55,56], an internationally standardized isolation method, which would be necessary for their clinical use, is lacking. Inherent to every method is the risk of co-isolating the other subtypes of EVs or lipoparticles, and the separation of different EV subtypes remains difficult. Further details are highlighted in the review “Extracellular vesicles in liquid biopsies” in this Special Issue. 

## 3. Clinical Applications 

Liquid biopsies have a great potential for initial tumor diagnosis, as well as for monitoring the response to therapy. Here, we elucidate the current applications and future perspectives of blood-plasma-based assays in lung cancer. An overview of studies, partly using liquid biopsy in mutational diagnostics, cited in this review can be found in the Appendix A.

### 3.1. ctDNA

The most robust data on all methods exist for the use and assessment of ctDNA, which has been integrated into the diagnostic work-up of NSCLC during the past decade. Current studies have shown the amount of ctDNA to correlate with the tumor stage (TNM), tumor burden, and metabolism [57,58,59]. Moreover, ctDNA assessment is suitable for monitoring therapeutic responses, as well as for detecting minimal residual disease (MRD). This would allow for the timely adjustments of individualized treatment, if needed. 

However, some key preanalytical considerations should be made ahead of ctDNA testing to improve the test performance. Most important is the identification of the correct patient cohort to be tested. As there is a variety of different tests available, each of them should be used carefully in the right clinical setting. The test requirements for a screening test of healthy individuals highly differ from the requirements for those of metastatic cancer to monitor the treatment response. In locally advanced and metastatic lung cancer, for example, state-of-the-art ctDNA tests focus on the evaluation of specific mutation sites, such as EGFR, ALK, and BRAF. However, upfront WGS might enable the detection of rare alterations and allow for an individualized treatment approach. A recent study evaluated the cost-effectiveness of a standard-of-care diagnostic approach vs. an upfront WGS for lung cancer patients [60]. If reimbursed and established, a liquid biopsy might allow for an easy and all-encompassing approach to evaluate for actionable mutations via ctDNA WGS. Another relevant state-of-the-art diagnostic focus in metastatic lung cancer is ctDNA testing for resistance genotyping under (targeted) therapy. Depending on the test method (e.g., targeted PCR-based assays vs. NGS), not only on-target resistance mutations, but also off-target resistance mutations can be detected (Table 2), thus implicating the importance of preanalytical considerations ahead of testing. Apart from that, a number of factors (e.g., disease status, tumor stage, histology, molecular pathology, and type of therapy, etc.) have an impact on the specific test performance, and should be considered carefully before choosing the right ctDNA test method.

#### 3.1.1. Mutational Assessment

Due to the rising number of targetable oncogenic drivers, the current NSCLC guidelines include molecular testing for already known driver mutations in regular diagnostic work-ups [11]. The diagnostic assessment should be performed at initial diagnosis and in cases of relapse or progression. Tissue analysis was formerly the accepted standard, but biopsies are often scarce, difficult to obtain, and may not represent the heterogeneity of the tumor. The use of liquid biopsies in the assessment of NSCLC is increasing, as they are easy to obtain and can possibly overcome interspatial/intratumoral variations. However, liquid biopsies should be obtained prior to the beginning of cytotoxic or targeted therapies, since effective therapy reduces DNA shedding and thus lowers the amount of ctDNA below a detectable threshold within 1–2 weeks after the therapy induction [61]. A recent analysis of the German CRISP register revealed that only 75% of all patients were tested for *EGFR* mutations. Most unsettling was that 12.4% of patients were not tested for any marker by any method [62,63], mostly because they were unable to obtain tissue biopsies and were unaware of the liquid biopsy. Here, by analyzing both tissue and liquid biopsy DNA samples, targetable oncogenic driver mutations were detected in 30–60% of patients [64,65]. Multiple meta-analyses have revealed the feasibility of assessing the mutational status via a liquid biopsy [66,67]. Moreover, the subsequent prospective trial validation led to the recommendation of liquid biopsies within current guidelines in Europe and the US, especially in cases with insufficient or unattainable tissue biopsies [68,69]. Recently, the International Association for the Study of Lung Cancer (IASLC) underlined the outstanding potential of liquid biopsies for the evaluation of oncogenic drivers and proposed that ctDNA-based mutational analysis be used preferentially. Especially in patients with progressive disease, they facilitate the choice of a specific therapy (“plasma first”) [23]. Moreover, the IASLC recommended their concurrent use in cases of questionable adequacy of the accessible specimens. The currently ongoing LIQUIK Trial (NCT04703153) aims to prove the non-inferiority of liquid biopsies compared to tissue biopsies in detecting the guideline-recommended molecular biomarkers, and might further foster the role of liquid biopsies in NSCLC diagnostics. 

#### 3.1.2. EGFR Mutations

ctDNA analyses initially entered the diagnostic field with the detection of *EGFR* mutations and the subsequent management of NSCLC treatment with tyrosine kinase inhibitors (TKI). In total, two PCR-based tests have been approved by the FDA and EMA, respectively, for detecting *EGFR* mutations. The cobas *EGFR* mutation test ^®^ detects 42 defined *EGFR* mutations with a high sensitivity [70], and allows a treatment selection from the companion drugs gefitinib and erlotinib or osimertinib. The therascreen^®^ *EGFR* kit [12] permits the initiation of treatment with afatinib, gefitinib, and dacomitinib in cases of finding an *EGFR* mutation. Comparative molecular analyses of liquid and tissue biopsies showed concordance in some trials, but often resulted in contradictory findings [70]. However, PCR-based methods can only detect a limited number of known *EGFR* mutations, which must be predefined before testing (c.f., comparison of PCR-based and NGS-based methods). Thus, both tests are unable to discover new *EGFR* mutations. Furthermore, PCR-based methods showed a high specificity (97% in a meta-analysis of nine studies) in *EGFR* detection, albeit with a low sensitivity of 51% [71]. In addition, high rates of false-negative results when detecting T790M mutations, a harbored resistance to first-generation TKIs, and only a 51% agreement to the matched tissue samples were seen with the cobas^®^ test in the AURA3 trial [72]. Hence, guidelines (e.g., ESMO practical guideline [73]) require a subsequent tissue biopsy in the case of a negative result from PCR-based liquid biopsies. Nevertheless, the less expensive PCR-based *EGFR* testing methods remain highly relevant as an initial step, especially for patients at risk of *EGFR* mutations. Still, unaddressed mutations can result in false-negative *EGFR* mutation test results, and a negative PCR-based test does not prove the absence of an *EGFR* mutation.

NGS-based approaches might overcome the latter disadvantage, as they allow for the simultaneous detection of all actionable targets, facilitate the detection of actionable variants, and thus permit rapid genotype-matched therapeutic decision making. Furthermore, NGS-based methods achieve a higher sensitivity and specificity [71,74], are therefore currently preferred over PCR-based methods, and are the first choice for liquid biopsy testing in the current ESMO guidelines for metastatic NSCLC. Thus, they might supersede PCR-based methods in the upcoming years. A total of two NGS-based methods were approved by the FDA in 2020: Guardant360^®^ CDx (Guardant Health, Palo Alto, CA, USA) and Foundation One^®^ Liquid Cdx (Roche, Basel, Switzerland). The Guardant360^®^ CDx-test, which allows for the use of osimertinib as a companion drug in the case of an *EGFR* mutation, showed a significantly increased number of molecular assessments compared to tissue-based standard diagnostics in a shorter time-period with a high (>98%) concordance between plasma and tissue [27]. The Foundation One^®^ Liquid Cdx test can assess over 300 gene alterations, allows the companion use of first- and second-generation TKI after the detection of sensitizing *EGFR* mutations, and is also approved for other cancers (e.g., breast and colon cancer). 

A liquid biopsy also allows for the early detection of disease progression, which is eventually seen in most patients undergoing TKI therapy as an expression of an acquired (secondary) resistance to *EGFR* TKI. There is an urgent need to evaluate the on- and off-target mechanisms of resistance (MOR), in order to better understand the treatment-induced tumor evolution and to adjust further therapies. The most common MOR is the *EGFR* T790M mutation, which can be effectively targeted with the third generation TKI osimertinib. The assessment of T790M with liquid biopsies is very well examined, and its detection in plasma could obviate the need for tissue biopsies, as it predicts an excellent response to osimertinib, similar to that observed in tumor genotyping [75,76]. Moreover, a subsequent ctDNA analysis in patients undergoing TKI treatment detected the T790M mutation in a median time of two months prior to the clinical progression, in 45% of all the patients monitored [77,78], allowing for the treatment to be adjusted before the clinical tumor progressed and the detriment of patient performance statuses were evident. The results from the FLAURA-trial pointed out that osimertinib significantly improved the survival of treatment-naive patients compared to the comparator-TKI, and the application of first-line osimertinib led to a lower frequency of T790M mutations [79]. Nonetheless, a resistance to osimertinib is emerging, and an evaluation of the distinct on- and off-target MOR to *EGFR* TKI via NGS-based liquid biopsies is urgently needed to effectively treat osimertinib-resistant NSCLC. This is the aim of the APPLE trial [80]. Beyond this, the MELROSE study (NCT03865511) addresses the question of whether there is an association between ctDNA levels and the occurrence of resistance to osimertinib or therapeutic escape, in order to choose the ideal treatment for the individual patient [81]. For instance, the promising results of the use of the *c-MET* inhibitor crizotinib in NSCLC cell lines harboring *MET* amplification, a known MOR to osimertinib, has shown a potential to overcome the acquired resistance [82,83]. 

In patients with T790M-negative plasma samples, a tumor biopsy is still warranted due to the possibility of false-negative results and a limited sensitivity of the liquid biopsy. In addition, other acquired *EGFR* mutations besides T790M should be examined, as some of them can be targeted by switching to first- and second-generation TKIs [84]. Interestingly, even when no other mutations are found and no tumor material can be assessed, a small cohort of patients might nonetheless profit from treatment with osimertinib. In the AURA1-trial, patients with tumor progression after TKI treatment, in whom neither the original *EGFR*-sensitizing mutation nor a T790M mutation could be detected with ctDNA, showed an improved outcome under osimertinib compared to those patients with proof of their sensitizing mutation (progression-free survival [PFS] of 15.2 months vs. 4.4 months). The negative test results for the T790M mutation were thus misleading in these cases. The lack of any detectable mutation in the plasma test might be explained by low DNA shedding, lower aggressiveness, and/or a lower disease burden in those patients.

#### 3.1.3. ALK Rearrangement

*ALK* rearrangement preferentially occurs in young patients, as well as in never-smokers. Fortunately, the treatment inhibition of downstream signaling via TKI is effective, irrespective of the, so far, 19 known fusion partners [85,86]. Where *ALK* fusions have not been effectively detected by PCR-based methods, an NGS-based liquid biopsy attains a high specificity (100%) and sensitivity (79.2%) [87]. In 2011, the multi-kinase-inhibiting drug crizotinib was the first FDA-approved substance for ALK-rearranged NSCLC, achieving response rates of 60% and disease control rates of 90% [88,89]. However, most patients ultimately progress while on crizotinib treatment [14]. Consequently, the current ESMO guideline [11] endorses the first-line treatment of ALK-driven NSCLC with a second-generation TKI, such as alectinib or its comparators brigatinib, ceritinib, or ensartinib [90,91,92]. Likewise, each of these substances is able to induce a renewed therapeutic response in up to 55% of patients with progression [93]. Other than these, lorlatinib is a potent third-generation *ALK*-TKI that has shown efficacy in patients without a response to one or more second-generation TKIs, and was recently EMA-approved, even as a first-line treatment [94]. However, the various TKIs have individual binding affinities and therefore a sensitivity to the *ALK* resistance mechanisms. Thus, even if the exact characterization of the mutations under treatment with *ALK*-TKI is not required for a therapeutic switch, this characterization via NGS panels, covering a variety of mutations, might help with finding the optimum choice of TKI for the patient. Most recently, the FDA approved alectinib as a companion drug for *ALK* rearrangement detected with the Foundation One^®^ Liquid Cdx test, since the first results of the ongoing BFAST trial not only underlined the suitability of NGS-based assays in detecting *ALK*-fusions, but also demonstrated that a subsequent treatment with alectinib yielded high overall response rates (ORR) [95]. NGS-based longitudinal monitoring also has prognostic value, as the detection of ctDNA (based on matched targeted NGS and shallow WGS) at the time of the initial diagnosis indicates a shorter time until progression, and the ctDNA levels might be highly elevated in some patients throughout TKI therapy until death [96,97].

#### 3.1.4. ROS1 Gene Fusions

Oncogenic *ROS1* gene fusions that induce the relevant activation of the *ROS-1* receptor tyrosine kinase can be detected with NGS-based liquid biopsies and subsequently targeted by crizotinib (with ORR of 73–93%) and entrectinib (ORR 41%, some cases of durable responses) in a first-line setting [98,99]. According to preliminary results, the consecutive ctDNA monitoring of a ROS1 rearrangement may help to predict the response to therapy [100], and allows for the early detection of resistance-mediating mutations in *ROS1* genes [101] that would ultimately require an adjustment of therapy.

#### 3.1.5. MET Rearrangements

The *MET* gene encodes for a receptor tyrosine kinase that, i.a., activates the signaling pathways involved in crucial cellular processes such as cell proliferation, survival, and growth [102]. Most prominently in NSCLC, *MET* exon 14 (*METex14*) skipping mutations are characterized by the fusion of exons 13 and 15, which ultimately impairs receptor degradation, thus overactivating the *MET*-mediated signaling [102]. *METex14* alterations are usually observed in the absence of other oncogenic driver mutations [102]. NGS-based Guardant360^®^ CDx and Foundation One^®^ Liquid Cdx tests are FDA-approved for the assessment of *METex14* skipping mutations in NSCLC. Encouraging results from the phase II GEOMETRY mono-1 trial led to the FDA approval of the *MET-*inhibiting oral agent capmatinib; patients with qPCR tissue-confirmed *METex14* skipping NSCLC, that was retrospectively assessed by Foundation One^®^ Liquid Cdx, achieved an ORR of 68% in first-line and 41% in second- or third-line treatment with capmatinib. More so, the study resulted in a 92% intracranial disease control and a 31% complete remission in patients with brain metastases [103]. Capmatinib was approved by the EMA in 2022. Most recently, the second-line use of tepotinib, another oral *MET* kinase inhibitor, was approved by the FDA and EMA, based on the results of the phase II VISION trial [104]. Here, *METex14* skipping was either determined by Guardant360^®^ CDx in plasma or with RNA from tissue specimens. Treatment with tepotinib induced an ORR rate of 43% over all therapeutic lines, with a median duration of response (mDOR) of approximately 11 months, and likewise demonstrated intracranial activity [105]. Furthermore, a high concordance of clinical response and decreasing ctDNA levels during treatment were observed [104,106]. Preliminary results of a second-line phase II trial (NCT02897479) with savolitinib indicated a mORR of 46%, while the mDOR was not reached in the interim analysis at 6.9 months [107].

#### 3.1.6. RET Rearrangements

*RET* rearrangements represent oncogenic drivers in young NSCLC patients with a non-smoking history. These patients are especially prone to brain metastases [108]. Several studies have shown liquid biopsies to be a reliable tool in identifying *RET* rearrangements [64,109,110]. The results of the LIBRETTO trial highlighted a high ORR after a *RET*-inhibiting selpercatinib administration in second- and first-line treatment (64% and 84%, respectively). While in both treatment lines, cases of durable response and intracranial activity [111] were observed, the approval was restricted to second-line treatment after platinum-based chemotherapy. With praseltinib being a highly effective treatment (an ORR of 70%, an mDOR not reached after 6 months, and the presence of CR in some cases), the ARROW trial [112] led to the approval of praseltinib in the first-line setting of *RET*-rearranged NSCLC. In line with other TKIs, the use of *RET* inhibitors is periodically limited, e.g., with acquired *RET* V480M gatekeeper resistance mutations [113] that are identifiable by a liquid biopsy [114]. With even greater impact, an analysis of over 32,000 samples showed that the finding of non-KIF5B-*RET* fusions contributed to *anti-EGFR* therapy resistance in cases with co-mutations [115].

#### 3.1.7. BRAF Mutations

*BRAF* mutations, and predominantly the V600E mutation, which accounts for over 50% of all cases and can reliably be detected in liquid biopsies [116], are associated with a dismal response to cytotoxic treatment [117]. In melanoma, in which *BRAF* mutations are more frequent, the BRAF inhibitors vemurafenib or dabrafenib have shown high response rates and have improved the survival of patients dramatically [118,119]. In parallel, the application of vemurafenib or dabrafenib alone, or with additional downstream *MEK* inhibition, has shown promising antitumor activity (an ORR of 33% for dabrafenib, 42% for vemurafenib, and 67% for dabrafenib/trametinib) in phase II trials for NSCLC [120]. Consequently, the EMA and FDA approved the combination therapy of dabrafenib and trametinib for *BRAF*-mutated, advanced NSCLC. 

#### 3.1.8. ERBB-2/Her-2 Alterations:

*ERBB-2/Her-2* alterations (overexpression, amplification, or point mutations) are common in breast cancer patients, where the monitoring of ctDNA under treatment has shown a high predictive value regarding the response to therapy [121]. *ERBB2/Her-2*-overexpressing NSCLC implies an inferior outcome [122], as *ERBB2/Her-2* amplification is known to be one of the acquired *EGFR* TKI resistance mechanisms [122] that can be identified by NGS-based liquid biopsies [123]. Recently, the *Her-2/neu*-targeting antibody-drug conjugate trastuzumab deruxtecan has shown an impressive ORR of 54% in heavily pretreated NSCLC patients [124]. 

#### 3.1.9. KRAS Mutations

Activating mutations in the *KRAS* gene are the most prevalent mutations in NSCLC, with the *KRAS* p.G12C variant being most frequent. The phase II CodeBreaK100 trial recently showed promising results for the use of sotorasib, a small molecule that irreversibly inhibits *KRAS* G12C, in pretreated *KRAS* p.G12C-mutated NSCLC [125]. Here, sotorasib exhibited an ORR of 37.1%, a disease control of 80.6%, and an mDOR of 11.1 months. However, the median PFS (6.8 months) and median overall survival (OS, 12.5 months) were still low in a cohort of pretreated, advanced NSCLC patients. Most recently, the first results of CodeBreaK200 phase III trial demonstrated that treatment with sotorasib significantly improved the disease control rate, as well as the PFS, and had a more favorable safety profile in a head-to-head comparison with docetaxel (NCT04303780), while OS data have not been provided yet [126]. The ongoing CodeBreaK101 trial (NCT04185883) is currently investigating the role of sotorasib in various combination therapies, including first-line settings. 

The feasibility of the NGS-based *KRAS* mutation assessment of liquid biopsies from NSCLC patients has already been proven, especially in cases with insufficient or unavailable tissue samples [127]. Other than NGS, the Idylla qPCR platform, that has a 100% concordance with NGS and a high sensitivity, may expedite cost-effective tools to assess the *KRAS* p.G12C mutation [128]. So far, the FDA has only approved the NGS-based Guardant360^®^ liquid biopsy CDx for tumor mutation profiling to identify *KRAS* p.G12C mutated patients with locally advanced or metastatic NSCLC, who may benefit from sotorasib.

#### 3.1.10. NTRK Fusions

Rare *NTRK* fusions can effectively be targeted by two FDA- and EMA-approved drugs (larotrectinib and entrectinib). Although the FDA approved the NGS-based FoundationOne^®^ CDx tissue test as a companion diagnostic to identify the fusions of *NTRK* genes in solid tumors, including NSCLC, so far, no NGS-based test for liquid biopsies has been approved. 

#### 3.1.11. Immunotherapy and Tumor Mutational Burden (TMB)

Immune checkpoint inhibitors (ICI), e.g., antibodies against PD-1 and PD-L1, have dramatically improved the outcome of advanced stage NSCLC patients. Independent of PD-L1 expression in tumor tissue, a combination of immunotherapy with platinum-based chemotherapy positively impacts the OS, more so than standard platinum-based chemotherapy alone, both in non-squamous [129] and squamous cell lung cancer [130]. However, NSCLC patients with a high tumor PD-L1 expression (i.e., their tumor proportion score [TPS]) and/or a high PD-L1 expression in tumor-infiltrating immune cells (i.e., their combined positive score [CPS]) benefit more from a single-agent immunotherapy compared to the previous standard-of-care platinum-based chemotherapy [131,132,133]. Unlike targeted therapies, ICIs reactivate cytotoxic T cells to overcome the tumor immune escape [134]. Although the tissue PD-L1 expression positively correlates with the treatment response, not all patients respond to immunotherapy, and to date, there is no alternative reliable cell-surface biomarker to predict the efficacy of ICI. 

The tumor mutational burden, which is usually assessed by whole exome sequencing (WES) or targeted NGS (the sequencing of cancer gene panels (CGP)) of tissue specimens (tTMB), refers to the total number of somatic coding mutations, base substitutions, short insertions, and deletions per tumor genome. A high TMB is positively correlated with an environment of a high tumor neoantigen load, which contributes to a lower PFS and OS [135]. Multiple retrospective trials have shown that a high tTMB (determined with a cut-off >20 mut/mb in the CHECKMATE-026 trial) predicts a greater benefit of the PD-1 inhibitors nivolumab or pembrolizumab, in both treatment-naive and pretreated NSCLC patients [136,137,138]. Furthermore, tTMB is non-overlapping with PD-L1 expression, as it constitutes another aspect of the immune phenotype, and thus enables the selection of ICI-treatable patients. Despite conflicting results regarding the relationship of the tTMB and ICI response in clinical trials [139,140], the determination of tTMB and the subsequent selection of patients for pembrolizumab treatment in advanced non-resectable or metastasized cancers, including NSCLC, was approved by the FDA in 2020 [141]. However, tTMB diagnosis is still limited to a minor proportion of NSCLC patients (34–59%), due to the mentioned difficulties in obtaining tissue specimens [142]. With established ctDNA panels or optimized gene panel algorithms for NGS-based blood TMB (bTMB) determination, a liquid biopsy may offer a valuable substitute for tissue. Here, bTMB has been demonstrated to have a positive concordance with tTMB [143,144], and was validated in several trials, confirming that a high bTMB predicts beneficial PFS and ORR when treated with immunotherapy [32,145]. bTMB can be measured precisely using targeted gene panels, but the accuracy is compromised when the bait size is less than 0.5 MB [139]. The FDA-approved tests, Guardant OMNI ^®^ (500 genes, 2.1 MB) and Foundation Medicine Liquid Cdx^®^ bTMB (394 genes, 1.14 MB), have been demonstrated to test for sufficiently large baits and to correlate with tissue [146]. In the MYSTIC phase III trial, ‘high’ bTMB was defined as a threshold of at least 20 mut/mb, assessed by the Guardant OMNI^®^ test [147]. High bTMB similarly predicted a clinical benefit of ICI therapy, with durvalumab and CTLA-4-inhibiting tremelimumab, over chemotherapy (a median OS of 21.9 months vs. 10 months, respectively). In addition, a cut-off concentration of ≥ 16 mut/mb measured with the Foundation Medicine^®^ bTMB test [145] identified ‘high’ bTMB in patients particularly prone to ICI [131,148]. This cut-off was defined in the POPLAR study and validated by the OAK trial [143,149]. It was likewise confirmed in the first randomized, prospective trial using bTMB (B-F1RST) as a predictor for the response to treatment with the PD-L1 inhibitor atezolizumab, showing a beneficial PFS and OS beyond the cut-off of ≥16 mut/mb [150]. 

Nonetheless, there is currently a high variability among techniques and their interpretations, and a lack of a unified threshold for “high TMB”. The preliminary findings of the TMB Harmonization Project [151], an initiative for standardization, recently demonstrated the feasibility of reference control lines to further align the estimation of TMB and identify targeted NGS assays, which must now be validated in prospective trials. Synoptically, the incorporation of both TMB and PD-L1 expression into multivariate predictive models [152] might be helpful in forecasting ICI response.

#### 3.1.12. Prognostic Value of DNA Methylation in cfDNA and CTCs

One of the major epigenetic modifications is DNA methylation. In tumorigenesis especially, the aberrant methylation of gene promoters might enhance or silence the transcription of RNA; hence, this directs a certain biological behavior of the malignant cell. When focusing on methylation analysis, one has to consider the different approaches of the obtained tissue and sample: while the liquid biopsy and analysis of cfDNA resulted in relatively high levels of APC_me_ and SLFN11_me_ sites in lung cancer patients, e.g., its corresponding CTCs exhibited a significantly lower frequency of promoter methylation [153]. In comparison to solid tissue, cfDNA and CTCs had fewer methylation signatures than tumor tissue itself, but cfDNA-detected alterations might derive from tumor adjacent tissue (non-ctDNA) [153]. Thus, before implementing the liquid biopsy analysis with regards to the methylation analysis, a predefined standardization and normalization of the tissue observed (e.g., cfDNA and CTCs in a liquid biopsy versus tumor tissue and adjacent tissue from tissue samples) has to be implemented and validly evaluated.

However, some early studies of cfDNA that evaluated DNA methylation analyses in cancer diagnostics allow a little glance at future perspectives. Here, Constâncio et al. detected a sensitivity of 64% and a specificity of 70% for a ‘PanCancer’ cfDNA analysis compiled of FOXA1_me_, RARβ2_me_, and RASSF1A_me_ in male early lung and prostate cancer patients. While positive results were also detected in 30% of the cfDNA healthy controls [154], its value as a potentially harmless method for cancer screening has to be verified by larger prospective trials.

With a focus on lung cancer patient diagnostics and staging, a liquid biopsy gathered APC_me_ and RASSF1A_me_ cfDNA methylation panel predicted the disease-specific mortality of lung cancer patients: compared to unmethylated sites, APC_me_ plus RASSF1A_me_ resulted in a Hazards Ratio of 3.9 (1.9–7.9) in lung cancer patients [154]. Such data could help to predict the cancer course and be useful for treatment decisions in the future.

Of current clinical relevance, TKI resistance mechanisms can be driven by DNA methylation. Here, a liquid biopsy cfDNA methylation analysis of prespecified sites (e.g., RASSF1A, RASSF10, APC, WIF-1, BRMS1, SLFN11, RARβ, SHISA3, and FOXA1) indicates the development of EGFR-TKI resistance in osimertinib-treated patients [155]. Likewise, a specific DNA methylation (i.e., 5-mC score) score was used to monitor the treatment efficacy and predict the disease progression in TKI-treated ALK-driven NSCLC patients with a liquid biopsy cfDNA analysis [156].

In conclusion, epigenetic changes such as DNA methylation might frequently be found in the liquid biopsies of cfDNA and CTCs. Still, adjacent tissue, tumor, and cfDNA analysis have to be normalized, and their deductive consequences must be well defined.

### 3.2. CTCs

CTCs provide parallel information on the mutational profile, CNVs, genomic rearrangements, and gene expression of a tumor. As they are released from the primary tumor site, as well as from metastases, the analysis of the CTCs might overcome the intratumor heterogeneity. Nevertheless, the CTC count does not correlate with the tumor burden, as only a fraction of NSCLCs shed CTCs [33], and the current data regarding the counting of CTCs and their correlation with the tumor stage are conflicting [157,158,159]. However, there is strong evidence that a high baseline CTC count predicts a poorer outcome in NSCLC patients, and is thus an independent prognostic factor in lung cancer [160]. Interestingly, monitoring the CTC count during therapy allows for an assessment of the disease development in real-time [33]. The maintenance of high CTC counts under therapy and during aftercare indicates a poorer prognosis, and identifies patients at risk of progression [161].

#### 3.2.1. Mutational Assessment

Even if mutational assessment is more established in the field of ctDNA, a molecular analysis of CTCs can provide additional information about the underlying driver mutations. Allele-specific PCR amplification in CTCs has been shown to confirm tissue *EGFR*-mutation in NSCLC in 11 of 12 patients (92%) [162]. Interestingly, in cases where no oncogenic drivers were found in the tissue-based examination, the assessment of CTCs was also able to identify *EGFR* mutations [163]. In addition to *EGFR* mutations, *ALK* rearrangements can be detected with a high sensitivity and high concordance with tissue [164] when analyzing CNVs, even though this is with a less broad range of mutations than with NGS-based methods in ctDNA [162,165,166]. The expression of *MET* in CTCs has been shown to correlate with their expression in the primary tumor tissues of NSCLC patients [167], expanding the number of molecular drivers that can be observed in CTCs. If a mutation is identified, CNVs can serve as a simple tool to monitor the genomic evolution of the tumor under TKI therapy, and predict the therapy resistance and clinical outcome [168].

#### 3.2.2. Immunotherapy and PD-L1 Expression

PD-L1 is co-expressed with EpCAM on >80% of CTCs from patients with metastatic lung cancer [169], making CTCs a suitable tool for identifying patients that qualify for ICI treatment. Interestingly, PD-L1 expression on CTCs in NSCLC patients is more often positive than on that in tissue, thus highlighting the potential use of CTC PD-L1 expression as a biomarker under immunotherapy [170,171]. Generally, the presence of PD-L1-positive (>1%) CTCs prior to treatment is associated with a poor outcome [170,172,173] in most trials. The persistence of PD-L1-positive CTCs after three and six months of ICI treatment was proposed as a predictor of mortality [174]. While conclusive proof is lacking, it is believed that the expression of PD-L1 on CTCs mediates immune escape [175], as patients with PD-L1-positive CTCs are more often non-responders to nivolumab [170]. More importantly, Nicolazzo et al. found PD-L1-positive CTCs in all patients that developed a resistance under ICI therapy [174], supporting the latter hypothesis. Nonetheless, further studies are needed to prove that correlation. 

Comparing PD-L1 expression on CTCs and in tumor tissue yielded widely divergent results: while Ilie et al. described a 93% concordance in their tissue-matched study [176], others saw no correlation [170], which might be explained by the different methods and/or antibodies used to analyze the PD-L1 expression in those trials [170,177,178]. This illustrates the need for standardizing the methods, and for future trials to validate the current knowledge and enlighten future perspectives on CTCs in NSCLC diagnostics.

### 3.3. miRNA 

miRNA profiles can reliably distinguish healthy controls from lung cancer patients [179,180,181]. Some profiles are specific for particular NSCLC entities [182,183,184], e.g., the detection of the highly specific hsa-miR-205 in plasma identifies a squamous cell carcinoma of the lung, with a high sensitivity and specificity [185]. In addition, the levels of blood-based miRNAs correlate with the disease stage and poor prognosis [186,187].

The analysis of miRNAs can also be used in early tumor stages for genotyping, monitoring TKI therapy [188], and selecting the patients that benefit from immunotherapy [189]. High expression levels of miR-195 and miR-122 were found to be associated with *EGFR*-mutant tumors, and were, moreover, independent predictors of survival [190]. In addition, increasing miR-21 levels were observed until there was a clinical progression under first-line TKI in *EGFR*-mutated NSCLC, anticipating TKI resistance and probably serving as MOR [191]. As miRNAs play a role in antitumor immunity by influencing the mRNA levels, and thus modulating the T-/NK-cell response [192] or PD-L1 expression [193], circulating miRNAs also have the potential to predict the response to ICI. Even though miRNAs have great potential, there is an urgent need for future trials to evaluate their role in disease progression, the evaluation of the therapeutic response, the prediction of progression, and the outcome of NSCLC patients.

### 3.4. EVs

The analysis of EVs has emerged as a method that can potentially complement the current gold standard of ctDNA analysis in the field of liquid biopsies for NSCLC (c.f. review “Extracellular vesicles in liquid biopsies” in this Special Issue). They are released from the primary tumor into the blood already at early tumor stages, are not as scarce as CTCs, and have the ability to pass the blood–brain barrier, thus allowing early access to tumors and metastases and their molecular compositions [194]. In NSCLC, EVs incorporate miRNAs, which are solely expressed in cancer but not in healthy subjects [195], and of which levels rise during tumor progression [196].

#### 3.4.1. sEVs

The role of sEVs and their potential usefulness in diagnostics, disease monitoring, and therapeutic stratification have been highly studied over the past years. Regarding their mutational assessment in NSCLC especially, the analysis of EV-derived miRNA has been proven to detect oncogenic drivers with a higher sensitivity than ctDNA [197], thus extending the therapeutic options of TKI therapy for more patients. Interestingly, the combined evaluation of resistance mechanisms in ctDNA and EV-derived miRNA showed a high sensitivity in lung cancer patients [198]. The miRNA analysis of sEVs can also offer prognostic information with regards to the response to osimertinib, and might thus be a screening tool regarding the individual choice of TKI [199,200]. 

Furthermore, PD-L1-positive EVs can inhibit the activation of CD8^+^ T cells. The finding of PD-L1^+^-EVs in plasma, and a rapid decline of EV levels under ICI treatment, seems to be highly prognostic with regards to an expectable response [201] and the outcome of NSCLC patients [202]. Furthermore, PD-L1 expression in sEVs is correlated with the progression, tumor burden, TNM stages, and metastatic capacity of NSCLC [203]. The results regarding the correlation between PD-L1 expression in EVs and the tissue remain controversial, ranging from strong [204] to no correlation [203]. EVs are also involved in immune escape mechanisms and the resistance to chemo- and immunotherapy, e.g., by distinct miRNA expression patterns or the overexpression of PD-L1 [55]. Thus, the assessment of sEVs might also help to predict the response to ICI or the failure of immunotherapy [201,205].

#### 3.4.2. lEVs

lEVs have long been known for their involvement in tumor progression and metastasis formation [55]. lEV levels are elevated in lung cancer patients [206], tend to be increased in late stage cancers [55,206], and predict a poor outcome [207], which demonstrates their potential as diagnostic and prognostic biomarkers in liquid biopsies. 

The combined analysis of EV-associated transcripts and ctDNA improved the sensitivity of EGFR mutational detection [208], with a high concordance with tissue biopsies [209]. As an alternative approach, an analysis of sEV mRNA levels to determine the oncogenic driver mutations of NSCLC has demonstrated a greater sensitivity than that of ctDNA assessment [197]. However, lEVs seem to reflect the mutational status even better than sEVs [210]. Furthermore, lEVs might be involved in the induction of resistance to both *EGFR* and *ALK* inhibitors via the trafficking of *EGFR* or *ALK*-mutated isoforms, and the subsequent activation of oncogenic pathways [211,212]. 

In synopsis, the monitoring of the presence of tumor-derived EVs and the investigation of the vesicle-specific content have dramatically emerged over the last years, and are a promising methodology for a rapid and accurate diagnosis, early tumor detection, and the guidance of therapy. At present, their clinical applications are hampered by a lack of standardization and multicenter studies, as well as a lack of efficient and cost-effective methods for the separation of EVs from non-EV lipid particles.

## 4. State-of-the-Art Approaches and Future Perspectives

### 4.1. Screening Programs

Since the clinical appearance of lung cancer is unspecific, the detection of tumors in early, potentially curable stages is challenging. With respect to liquid biopsies, little DNA is shed from small tumor masses and ctDNA concentrations are very low, even prior to treatment [23]. Thus, the low sensitivity of detecting ctDNA in early-stage lung cancer is a major challenge. The sensitivity has been increased by the use of NGS-panels [213] or targeted error sequencing (TEC-Seq) [214]. Several programs have fused ctDNA diagnostics with other parameters, and by integrating distinct molecular features, the lung cancer likelihood in the plasma (Lung-CLIP) algorithm was able to robustly discriminate cancer patients from the risk-matched controls [213]. The multi-analytic blood test CancerSEEK combines ctDNA detection with known protein biomarkers of early cancer stages [215,216]. As early cancer detection ultimately reduces the number of cancer deaths, these multi-analytic blood tests (such as CancerSEEK) have shown promising results for cancer screening. However, ongoing trials aim to increase the clinical sensitivity of those tests. Similar to the implementation of LD-CT scans as a screening method for lung cancer, the use of such multi-analytic blood tests should be limited to patients at risk for cancer, in order to increase the sensitivity. In lung cancer, factors like the patient’s smoking status, their history of exposure to carcinogens, and their age should be considered when selecting the most suitable group of patients for blood screening. Additionally, the prevalence of cancer should be taken into account in order to increase sensitivity [217]. However, CancerSEEK has been proven to be highly specific, e.g., only 7 of 812 healthy individuals tested positive [215]. Thus, it might even be possible in future to screen not only patients at risk, but also healthy individuals for cancer.

Prospective trials, such as the DETECT-A trial [218], the SUMMIT trial (NCT03934866), or the ASCEND trial (NCT04213326) will provide new insights into a risk-stratified approach based on the results of the liquid biopsies (Appendix A). 

Moreover, with few exceptions, such as Crohn disease [219], endometriosis [220], or pregnancy [221], CTCs might enable an early tumor detection in lung cancer with a high specificity. The detection of CTCs in patients with CT-confirmed suspicious pulmonary nodules reliably discriminates malignant from benign lesions [222]. More importantly, “sentinel” CTC detection in COPD patients identified those patients who developed cancer in the follow-up period [223]. The ongoing multi-center cohort AIR-trial [224] aims to prove whether the image-based monitoring of “sentinel” CTC-positive COPD patients allows for the early diagnosis of lung cancer and improves prognosis (Appendix A). 

### 4.2. Monitoring MRD after Curative Intended Therapy and Surgery

ctDNA and CTCs in liquid biopsies are also being investigated for their use as biomarkers to determine the MRD levels and monitor the follow-up after curative intended therapy and surgery.

In a cohort of 41 NSCLC patients, 91.7% (22) of the pre-op-identified 24 plasma ctDNA mutations had a decrease in their mutation frequency within only two days after surgical tumor resection. Moreover, the presence of ctDNA had a higher positive predictive value of residual disease than that of six tumor biomarkers [225]. The ongoing randomized and controlled phase III SUPE_R trial (NCT03740126) is currently investigating the efficacy of ctDNA analysis using serial liquid biopsies every three months in stage I-III NSCLC patients after curative intended treatment [226]. In addition, the observational *ORACLE* (NCT05059444) study aims at demonstrating the ability of a novel ctDNA assay to detect recurrence in individuals treated for early-stage solid tumors.

Newman et al. developed a method called CAncer Personalized Profiling by deep Sequencing (CAPP-Seq) to identify cancer-specific genetic aberrations in ctDNA analysis [227]. In 94% of patients with tumor recurrence after the curative intended treatment for stage I-III lung cancer, ctDNA detection using CAPP-Seq was positive in the first post-treatment blood sample, indicating the reliable identification of MRD [228]. Post-treatment ctDNA detection preceded radiographic progression in 72% of patients by a median of 5.2 months [228]. Likewise, in the ongoing TRACERx (NCT01888601) study, pre- and post-surgical plasma ctDNA profiling is performed, blinded to relapse statuses to assess the clonal tumor evolution and tumor heterogeneity, and the mutational changes under therapy. The ctDNA profiling used in TRACERx was able to identify 93% (*n* = 13) of those patients with a confirmed relapse, prior to or at clinical relapse. The median interval between the ctDNA detection and the NSCLC relapse confirmed by a CT scan was 70 days [229]. Interestingly, it was shown that maintenance with immunotherapy in MRD-positive patients after curative intended therapy provided better outcomes [230]. 

Although patients with detectable ctDNA after surgery have a significantly lower PFS and OS than those with undetectable ctDNA after surgery [228], there are still some patients with undetectable ctDNA that ultimately recur [231]. Thus, interpreting the absence of ctDNA remains difficult. 

Although the preliminary data for using ctDNA after surgery to identify the patients at risk of relapse appear promising, further data, including the awaited MRD data from the ADAURA trial [232], are required to elucidate which patients would benefit from an adjuvant therapy such as osimertinib (EGFR-TKI). 

### 4.3. Treatment Monitoring in Advanced Stage (Metastatic) NSCLC

ctDNA detection can also be used to monitor the treatment response in advanced (metastatic) tumor stages. As one knows that the mutation load is correlated with the survival rate [233], the ctDNA-based monitoring of *EGFR* mutations might allow for an early prediction of the resistance to EGFR-TKI in NSCLC patients [234]. The FASTACT-2 study found that the changes in the cfDNA *EGFR* mutation status might predict clinical outcomes, even prior to routine CT scans [235]. In patients with *EGFR* mut^+^ at baseline, the median PFS and median OS were shorter if the patients were still EGFR mut^+^ at cycle 3, than if they were EGFR mut^−^ at that point (7.2 vs. 12.0 months and 18.2 vs. 31.9 months, respectively.) [24].

The prognostic value of (i) the detection of the EGFR mutation in the plasma at baseline, as well as of (ii) the clearance of the EGFR mutated ctDNA under systemic therapy, was confirmed by several other studies, including the FLAURA trial [235,236,237,238]. The ongoing phase II APPLE trial (NCT02856893) on EGFR mutant, advanced NSCLC patients, compares the sequential T790M test using a ctDNA analysis of liquid biopsies to conventional radiological procedures, with regards to the prediction tumor progression [80]. 

Confirming the therapeutic response to ICI with imaging methods is challenging due to the induction of inflammatory processes with leukocyte infiltration that can be misinterpreted (pseudoprogression) in up to 6% of patients [239]. The subsequent monitoring of ctDNA levels via targeted NGS-based methods can predict the response to ICI [240]: After an initial peak due to the massive induction of apoptosis, a rapid decrease in the ctDNA levels under treatment, especially in the first four to eight weeks, correlates with a response and favorable outcome [241,242], while patients without a molecular response had a shorter PFS and OS compared to molecular responders [243]. There is also evidence from a small patient cohort that the increase in ctDNA might identify ICI non-responders prior to imaging [244]. The now-recruiting LIBERTYLUNG trial (NCT04790682) prospectively examines the suitability of ctDNA to predict the response to the PD-L1 inhibitor pembrolizumab (+/− chemotherapy) in treatment-naive metastatic NSCLC patients, with at least one detectable mutation. However, further studies are crucial to standardize the diagnostic thresholds and refine the methodology of liquid biopsies in this regard.

### 4.4. Limitations

Despite the great potential for the use of liquid biopsies in diagnostic and therapeutic decision making for patients with lung cancer, there are still several limitations of the current applications. 

#### 4.4.1. Sensitivity and Specificity 

Tumor tissue genotyping is generally associated with a higher sensitivity than liquid-biopsy-based genotyping [14,23]. The sensitivity of ctDNA analysis mainly depends on the amount of ctDNA shed by the tumor; hence, its stage and type. The sensitivity is increased in patients with extra-thoracic metastases (especially bone and liver metastases), [245] as well as in patients with a high tumor burden [246]. However, 20% of stage IV NSCLC patients do not shed ctDNA [62]. Currently, highly sensitive assays can achieve sensitivity levels of up to 85% in advanced stage lung cancer [14], which have been further improved with the use of deep sequencing methods. Although a positive result in ctDNA analysis using a validated test is usually sufficient to initiate targeted therapy [14], a negative result should be further clarified, e.g., with a tissue biopsy, due to the high risk of false-negative results, especially in slow-growing or early-stage tumors [71,247]. 

Although the evolvement of deep sequencing methods such as NGS has increased sensitivity, it brings the inherent risk of false-positive results—variants with low allele frequencies especially can result in a single false-positive result and impact the data interpretation. However, some technical adaptations have been developed to further improve specificity (e.g., molecular barcoding or error-proofing algorithms). 

False-positive results can also be caused by germline variants or the clonal hematopoiesis of indeterminate potentials (CHIP) that contribute to the bulk of cfDNA, and can interfere with the interpretation of ctDNA analysis results [247,248,249]. 

Some methods have been developed to avoid the false-positive detection of mutations caused by CHIP, e.g., a combined screening of matched white blood cells and ctDNA [250,251]. Moreover, a machine learning method, which can discriminate clonal hematopoiesis mutations from frequently recurring genetic alterations in NSCLC, has been validated [213,247]. It must be noted that conditions unrelated to cancer, such as infections, trauma, or inflammations, can increase the cfDNA levels in blood and might also lead to false-positive results [252].

#### 4.4.2. Early Stages

Molecular testing in the early stages of NSCLC is not as widely reflected in current guidelines as it is for advanced stages. However, it has gained importance in the adjustment and individualization of therapy. The ADAURA trial demonstrated that adjuvant treatment with osimertinib in stage IB-IIIA EGFR-mutated NSCLC significantly improved the disease-free survival, thus leading to the approval of osimertinib in Ex19del or L858R EGFR-mutated NSCLC [232]. Consequently, testing for *EGFR* mutations, which were found in 30% of stage I-III NSCLCs in a meta-analysis [253], is now recommended in the current National Comprehensive Cancer Network (NCCN) guidelines, to determine whether a patient might benefit from adjuvant treatment with osimertinib [254]. CTCs can also be detected in the early NSCLC stages. Crosbie et al. outlined the large potential of CTCs for post-surgical risk stratification [255]. In their study, in 33 patients with NSCLC stage IA-IIIA, they found that the detection of tumor micro-emboli or ≥2 CTCs in the pulmonary vein draining the tumor was associated with an eight-fold increase in the risk of disease recurrence, and a seven-fold increased risk of death. Hence, the CTCs detected during surgery might identify the patients for whom an adjuvant therapy during early-stage disease would prevent the disease recurrence.

#### 4.4.3. Financial Aspects/Accessibility

As more patients are given targeted therapies, the outcome of the patients suffering from NSCLC has improved. An analysis in Canada showed that therapeutic treatments yielded a gain of 168 life years, but at a cost of 14.7 million US dollars [256]. Even if the cost of an initial broad NGS-based mutational assessment (e.g., using liquid biopsies) is now comparable to that of sequential testing or hotspot panels [257], standard-of-care tissue testing is significantly less expensive [23]. In addition, a reimbursement from national health systems or health insurance is lacking in many countries, e.g., in Spain. NGS-based methods for analyzing liquid biopsies require the newest equipment that is generally only available in First World countries, which increases the global social gap and disparity, and also hinders their broad clinical introduction.

## 5. Conclusions

As a result of intensified research activity, the scope of the application of liquid biopsies is currently expanding. The new NGS-based techniques have enhanced the ability to analyze ctDNA, EVs, or CTCs, in order to assess an increasing number of accessible targets. Liquid biopsies, and ctDNA in particular, have already secured their position in clinical routine, where they are useful for the risk stratification of NSCLC patients, as well as for monitoring therapeutic efficacy, although further elucidation of the pathophysiological role and the implications of CTCs and EVs is required. 

However, there are several obstacles limiting the role of this technique. The absence of standardized protocols for extracting and analyzing the DNA, and interpreting the data, together with the low amount of targeted biomarkers in plasma, are impeding the further integration of liquid biopsies into diagnostic routine. Additionally, most studies in this field are based on small sample sizes, which gives them an inferior level of evidence. Hence, further large-scale, prospective, randomized, and controlled trials are needed to clarify the role of liquid biopsies, to extend the current knowledge, and to strengthen the weight of liquid biopsies in diagnostics, risk stratification, the assessment of individualized therapy, disease monitoring, and, ultimately, in achieving better outcomes for NSCLC patients.

## Figures and Tables

**Table 2 cancers-15-01430-t002:** Comparison of targeted PCR-based assays and NGS for DNA analysis (adopted from [9,12,22]).

	Method	Limit of Detection	Advantages	Disadvantages	Clinical Use
Targeted PCR-Based Assays					
Digital PCR (ddPCR, BEAMing)	DNA fractionation into different reactions sites for parallel qPCR	0.04–0.1%	Highly sensitive and specific Quantitative Low turnaround time	Not suitable for unknown alterations	Resistance genotyping
qPCR (e.g., Cobas^®^, Therascreen^®^)	Amplification of predefined DNA sequences	0.1–1%	Highly sensitive and specific Low turnaround time	Limited multiplexing Semi-quantitative	Initial and resistance genotyping of known mutations
Next generation sequencing (NGS) methods					
WGS	NGS of the full genome	10%	Detection of unknown alterations and new mechanisms of resistance (MOR)	Low specificity (false positives) Risk of detecting germline mutations Low sensitivity extensive bioinformatics High costs	Not in clinical routine, more experimental use
WES	NGS of the full exome (i.e., coding regions)	5%			
Hybrid-capture based NGS (e.g., Guardant360^®^ CDx, FoundationOne^®^ Liquid CDx)	Sequencing of target regions, that are captured by hybridization	0.001–0.5%	High sensitivity detection of SNPs, CNVs and gene fusions Simultaneous detection of predefined genes of interest as well as unknown mutations	Lower specificity (65%) than amplicon-based NGS unable to detect fusions without prior knowledge of partners	Initial and resistance genotyping
Amplicon-based (PCR-capture) NGS	Sequencing of target regions, that are amplified by PCR	0.01–2%	High sensitivity detection of SNPs, CNVs and gene fusions Simultaneous detection of predefined genes of interest as well as unknown mutations Higher specificity than hybrid-capture NGS (>99%)	Bias of CNVs and AFs (due to amplification) Unable to detect fusions without prior knowledge of partners	

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
