# Peer review of "Liquid Biopsies in Lung Cancer"

_cancers, 2023, doi:10.3390/cancers15051430_

Round 1

Reviewer 1 Report

In this review, the authors focused their attention on liquid biopsy and its potential clinical applications in NSCLC patients, including cancer diagnosis, treatment plan prioritization, minimal residual disease detection, and dynamic monitoring of the response to cancer treatment. Most studies actually have indicated that ctDNA testing is very important in diagnosing NSCLC, predicting clinical outcomes, and response to targeted therapies and immunotherapies, and detecting cancer recurrence. It is very interesting the discussion about the potential role of liquid biopsy in improving early lung cancer detection. Kemper M and colleagues resumed the current applications of ctDNA for the management of NSCLC and discussed future directions for guiding clinicians in treatment decision-making.

To my eyes, the paper is good by considering the large number of studies proposed, ongoing clinical trials that actively investigate the role of ctDNA and CTCs and their mutational assessment in the treatment of NSCLC patients. The scientific content seems good as well as the English style and language used in the manuscript need. Moreover, I appreciate the scientific efforts to organize this paper and I think that the rationale is well-discussed. The resolution of the table and the reference list meets the quality requirements of our journal. Moreover, the results were clearly discussed and corroborated with what is shown. Finally, the data analyses were interpreted in a comprehensible manner. I didn’t observe any remarkable incongruences throughout the text.

In my opinion, this article should be considered with minor revisions in light of comments/suggestions as indicated below.    

1-      Since the liquid biopsy is a rapidly evolving diagnostic tool for precision oncology that has recently found its way into routine practice as an adjunct to tissue biopsy, I think that it could be important to consider pragmatic scenarios for the use of ctDNA from blood plasma to identify actionable targets for therapy selection in NSCLC patients. It is well established that different clinical scenarios may require different analysis strategies but key preanalytical considerations for ctDNA mutation testing in NSCLC could probably impact the performance. I would ask the authors to integrate the preanalytical steps of ctDNA testing and explain how different many factors can affect it.

2-      Such studies involving research into multianalyte blood tests for various cancer types are conceptually and practically being explored with promising results regarding ultrasensitive testing (e.g. CancerSEEK) to enhance early detection by liquid biopsy screening. What’s your consideration about it? And how could increase clinical sensitivity in the future? Please discuss this point.

3-      As one of the intensely studied epigenetic modifications, DNA methylation plays a key role in various diseases. Remarkably, DNA methylation usually occurs in the very early stage of lung cancer. Considering the critical role of DNA methylation in tumorigenesis and cancer screening, how to detect DNA methylation status accurately and quickly is particularly important. It would be interesting if the authors will elaborate on the technologies for genome-wide DNA methylation analysis and specify the prognostic value of DNA methylation in cfDNA and CTCs.

4-      Please recheck the reference list, fix some abbreviations and punctuation throughout the text.

Reviewer 2 Report

Indeed, there is currently an urgent need for new therapeutic and diagnostic approaches to detect tumors at an early stage and monitor response to therapy. In addition to the well-established tissue biopsy assay, liquid biopsy-based assays can become an important diagnostic tool. The analysis of circulating tumor DNA (ctDNA) is the most recognized method, followed by other methods such as analysis of circulating tumor cells (CTCs), miRNAs (miRNAs) and extracellular vesicles (EVs). PCR and NGS based assays are used to evaluate mutations in lung cancer, including the most common driver mutations. However, cDNA analysis may also play a role in monitoring the effectiveness of immunotherapy and its recent advances in modern lung cancer therapy. The authors reviewed the main methodological aspects of liquid biopsy, diagnostic and therapeutic possibilities, limitations in terms of sensitivity (risk of false negative results) and specificity (interpretation of false positive results).

I liked the article, it quite fully reflects the current state of research in this area and should undoubtedly be published.

1. I would like to see in section 3 a summary table that would summarize all the studies, mentioning the number of patients, cancer stage, histological type and diagnostic result.

2. In the methodological part, it would be nice to add a methodology for selecting studies to be included in this review and an analysis of the number of publications in each area.
